# Peatland organic matter quality varies with latitude as suggested by combination of FTIR and Ramped Pyrolysis Oxidation

Katy J. Sparrow[1,2]*, Jeffrey P. Chanton[1], Ulrich M. Hanke[3,4], Mark D. Kurz[3,4], Ann P. McNichol[5]

1 Department of Earth, Ocean, and Atmospheric Science, Florida State University, Tallahassee, FL, United States of America, 2 Department of Geosciences, Georgia State University, Atlanta, GA, United States of America, 3 NOSAMS Laboratory, Geology and Geophysics, Woods Hole Oceanographic Institution, Woods Hole, MA, United States of America, 4 Marine Chemistry and Geochemistry, Woods Hole Oceanographic Institution, Woods Hole, MA, United States of America, 5 Geology and Geophysics, Woods Hole Oceanographic Institution, Woods Hole, MA, United States of America

* ksparrow@gsu.edu

**Data Availability Statement:** All relevant data are available for download at https://doi.org/10.5281/zenodo.7039662.

## Abstract

We employed two compelling and distinct methods, Fourier Transform Infrared Spectroscopy (FTIR) and Ramped Pyrolysis Oxidation (Ramped PyrOx), to examine the quality of organic matter (OM) stored in four peatlands located along a latitudinal gradient (Tropical (4˚N), Subtropical (27˚N), Boreal (48˚N), and Polar (68˚N)). FTIR was used to quantify the relative abundance of carbohydrates, a relatively labile compound class, and aromatics, which are more recalcitrant, in a sample set of four peat cores. These samples were then prepared using Ramped PyrOx, a second, independent method of determining OM quality that mimics the natural diagenetic maturation of OM that would take place over long time-scales. Previous large-scale studies using FTIR to evaluate OM quality have observed that it generally increases with increasing latitude (more carbohydrates, less aromatics). Here, we demonstrate that the Ramped PyrOx approach both validates and complements the FTIR approach. The data stemming from each Ramped PyrOx preparation was input to a model that generates an estimated probability density function of the activation energy ($E$) required to break the C bonds in the sample. We separated these functions into three fractions ("low $E$," "medium $E$," and "high $E$") to create Ramped PyrOx variables that could be quantitatively compared to the compound class abundance data from FTIR. In assessing the agreement between the two methods, we found three significant relationships between Ramped PyrOx and FTIR variables. Low $E$ fractions and carbohydrate content were positively correlated ($R^2 = 0.51$) while low $E$ fractions were negatively correlated with aromatic content ($R^2 = 0.58$). Medium $E$ fractions were found to be positively correlated with aromatics ($R^2 = 0.69$).

**Funding:** KJS was supported by a Diversity Postdoctoral Scholar Award from the Provost's Office at Florida State University (FSU). KJS and JPC were awarded a Radiocarbon Research Initiative by the National Ocean Sciences Accelerator Mass Spectrometer Facility (NOSAMS). UMH was supported by the National Science Foundation (NSF) during his postdoctoral position at NOSAMS (OCE 1755125). Neither the Provost's Office at FSU nor NSF had a role in study design, data collection and analysis, decision to publish, or preparation of the manuscript. Laboratory preparations and analyses were performed at NOSAMS by the authors. Two coauthors (UMH and MDK) are affiliated with NOSAMS."

**Competing interests:** The authors have declared that no competing interests exist.

## Introduction

Peatlands are forested or non-forested wetlands characterized by soils rich in organic matter (OM). Throughout the Holocene, new peatlands formed in low, middle, and high latitudes [1–3] and the amount of carbon (C) stored in global peatlands increased significantly during this period, to the estimated 530 ± 160 Pg C stored today [4]. The net increase in peatland C stocks during the Holocene was due to northern peatland expansion (>40˚N) while the amount of C stored in lower latitude peatlands has been relatively constant for the last 130,000 yr [2]. Although the rates of primary production in peatlands exceed those of decomposition, allowing for peatlands to act as net C sinks, anthropogenic warming and disturbance are projected to increase peatland greenhouse gas emissions and possibly turn these ecosystems into a net source of C [4–9]; this climate feedback remains to be observationally detected in unmanipulated environments. OM decomposition and C loss are projected to accelerate due to more frequent and/or prolonged droughts with climate change, as oxygen entering drying soils gives rise to the enzymes needed to decompose peat [10–13]. Warming also slows the growth and production of *Sphagnum* mosses [14], possibly as they become shaded by the growth of shrubs [15]. Loss or reduction of the *Sphagnum* community would diminish both a major building block of peatlands worldwide as well as their resistance to decomposition, as *Sphagnum* produces phenolic compounds, which slow the decomposition of OM [16, 17].

Peatlands, globally, have a range of differences that will affect what C losses will result from warming and disturbance, for example, the lability/degradability of OM, which we refer to as "OM quality," hydrology, presence of permafrost, and vegetation [9, 18], all of which are related to the master variable of climate. The existence of 100s Pg C in tropical peatlands [19] is still somewhat enigmatic and perhaps antithetical to the premise that peatland C in high latitudes will not be preserved as climate change proceeds. OM preservation mechanisms are a key consideration in understanding peatlands and their vulnerability (or lack thereof) to decomposition. As peatlands are wetlands and thus are waterlogged year-round or seasonally, the presence of high-water tables and their control on redox conditions is a preservation mechanism that is present in all peatlands, to varying degrees. Cold climate and ice are additional physical preservation mechanisms of peatland C, with permafrost peatlands, in which OM is trapped in a matrix of ice, representing ca. 35% of C stored in global peatlands [4]. Rather than be inherently vulnerable to warming climate, permafrost, itself, may also act as buffer to warming and attendant decomposition [20]. Using a coupled model of long-term peat dynamics under emissions pathway RCP8.5, Treat et al. [20] found that permafrost peatland sites had less C loss by 2100 than sites without permafrost, despite accelerated warming at the permafrost sites. Other preservatives are biochemical, for example, the aforementioned phenolic compounds in *Sphagnum*-containing peatlands [17, 21], or physiochemical, such as OM interactions with minerals [22]. Yet, minerals (quartz, clays, etc.) are not common within peat soil matrices, so while minerals have been shown to effectively protect OM from degradation in many soils and sediments [22, 23], minerals are unlikely to be a significant preservative of global peatland C.

Fourier Transform Infrared Spectroscopy (FTIR) has been used for over three decades to characterize peat OM [24–30] and the advancements in FTIR data processing by Hodgkins et al. [27] allow for rigorous, quantitative comparisons of the estimated carbohydrate, aromatic, and aliphatic content across samples. Using FTIR, Hodgkins et al. [27] observed a latitudinal gradient in peat composition from 14 cores across seven sites, such that high latitude peat was more enriched in carbohydrates (O-alkyl C) while peat at lower latitudes exhibited a greater content of aromatic moieties (Klason lignin) [31]. Two main drivers of this latitudinal trend were proposed, the first being that the litter input to the soils in warmer climates generally has a higher content of recalcitrant aromatics and lower content of carbohydrates

compared to plants in colder climates [27]. Secondly, the authors suggested that more extensive humification and degradation processes operating in warmer, low latitude climates preferentially removes labile carbohydrates, thus enriching the remaining soils in aromatics. This compositional change, in turn, serves as a negative feedback to further OM decomposition in (sub)tropical climates. These findings were then confirmed, and the role of the vegetation driver diminished, in a subsequent study with a much larger dataset including over 1,000 samples from 165 peatlands spanning from 79˚N to 65˚S degrees latitude [30]. In the study by Verbeke et al. [30], data analysis indicated that variations in peat carbohydrates and aromatic content were more attributed to three variables that determine climate (distance from the equator, mean annual temperature, and elevation) than they were to variations in peat-forming surface vegetation. The authors concluded that while a large portion of the labile content in high latitude peatlands would be lost upon warming, a residual fraction of peat would likely survive and become more aromatic-rich and more resistant to rapid decomposition, eventually slowing the peatland carbon climate feedback.

It has been shown that carbohydrate content is a stronger predictor of soil OM lability relative to elemental ratios (e.g. Carbon to Nitrogen (N), C:N), pH, or OM $^{14}$C content [32, 33]. These cellulose-derived compounds have a relatively high C oxidation state, which has been associated with greater reactivity in both empirical and theoretical studies [18, 34–38]. *Sphagnum*, which is particularly abundant at high latitudes, can inhibit the decomposition of OM, including carbohydrates, by releasing phenolic compounds and organic acids [39–42]. Lignin, an aromatic-based structure, is relatively stable in anaerobic soils [43–45]. Interestingly, the latitudinal gradient observed by Hodgkins et al. [27] and Verbeke et al. [30] are similar to the downcore trend in many peat cores that show the degradation of carbohydrate compounds and resulting enrichment of aromatic compounds with depth [38, 42].

In this study, we sought to gain more insight into the compositional variability of peatlands by applying a unique suite of geochemical analyses to peat cores spanning 64 degrees of latitude (**Table 1**), which ensures a large range of potential variability. We chose ramped pyrolysis oxidation (Ramped PyrOx), a serial thermal oxidation/pyrolysis technique that mimics the natural diagenetic maturation of OM and allows for thermally separating a sample for C isotope analyses [46, 47], to complement FTIR and elemental analysis (C:N). To our knowledge, this is the first study to perform side-by-side FTIR and Ramped PyrOx analyses in relation to C:N analyses, a more conventional method of assessing the degree of OM humification. We aimed to apply this trio of analyses to a small subset of the peat cores examined in the much larger studies of Hodgkins et al. [27] and Verbeke et al. [30] that utilized FTIR. Following up on those investigations [27, 30], we sought confirmation that differences in OM quality exist between warm, low latitude peatlands and cooler, high latitudes sites where the bulk of the peat C storehouse lies [4].

We tested the hypothesis that the data derived from Ramped PyrOx would correlate well with FTIR data (H1), confirming the validity of conclusions on peatland OM quality made using either approach. We expected to find that the latitudinal gradient of apparent greater OM quality at high latitudes, previously observed in larger studies using the FTIR approach [27, 30], would be observed with the Ramped PyrOx technique. As well, in this study, we tested the hypothesis that differently aged fractions could be thermally separated within individual peat samples using the Ramped PyrOx technique (H2).

## Materials and methods

### Sites and sampling

Peat samples were collected from four sites with different climates, which we will refer to herein as Tropical, Subtropical, Boreal, and Polar. No permits were required to access or

**Table 1. Characteristics and locations of peat core sampling along a latitudinal transect.**

| Climate Zone | Location | Lat. (°N) | Mean Annual Temp. (°C) | Mean Annual Precip. (cm) | Peat Core ID | Coring Location; Water Table Depth | Depth horizons sampled (cm) |
|---|---|---|---|---|---|---|---|
| Tropical | Ulu Mendaram Conservation Area, Brunei | 4.4 | 27 | 291 | MDM11-2A | forested peat dome; -20 to +20 cm | 0–1 |
| | | | | | | | 30–31 |
| | | | | | | | 40–41 |
| | | | | | | | 190–191 |
| Subtropical | Loxahatchee NWR, FL, USA | 27 | 24 | 138 | Lox3 | inundated peat marsh; 50 to +100 cm | 1–2 |
| | | | | | | | 30–31 |
| | | | | | | | 70–71 |
| | | | | | | | 180–181 |
| Boreal | Marcell Experimental Forest, MN, USA | 48 | 4.2 | 78 | T3F | hollow in peat bog; -10 to 0 cm | 0–10 |
| | | | | | | | 20–30 |
| | | | | | | | 40–50 |
| | | | | | | | 100–125 |
| Polar | Stordalen Mire, Sweden | 68 | -0.5 | 101 | CPP | rim of collapsed permafrost palsa; dry with active layer base ~60 cm depth | 1–5 |
| | | | | | | | 30–35 |
| | | | | | | | 50–55 |
| | | | | | | | 70–75 |

perform work in the field sites. Four depth horizons were analyzed at each of the four sites, for a total of 16 samples. **Table 1** summarizes key characteristics of each peat core. The cores sampled here are a small subset of the cores sampled by Hodgkins et al. [27] and Verbeke et al. [30]; these prior studies analyzed multiple cores at each site using FTIR. Hodgkins et al. [27] analyzed two cores at our Tropical site, two cores at our Subtropical site, two cores at our Boreal site, and one core at our Polar site. Verbeke et al. [30] analyzed eight cores at our Tropical site, no cores at our Subtropical site, but three cores located within one degree of latitude south of it at another Everglades location, 21 cores at our Boreal site, and eight cores at our Polar site.

The Tropical peat samples were collected from a core obtained in November 2011, "MDM11-2A," (4.3727°N, 114.3550°E) from a pristine ombrotrophic peat swamp forest located in the Ulu Mendaram Conservation Area in the Belait District of Brunei Darussalem, northwest Borneo; all sampling details and the physical and chemical properties of this core are described by Dommain et al. [48]. *Shorea albida* trees and an understory of densely populated *Pandanus andersonii* characterize this site and its peat is composed of woody debris, leaves, and non-woody plants [27]. The water table ranges in depth from 20 cm below the surface to 20 cm above the surface in small pools [27, 48]. Dean et al. [49] report that peatland pools are dynamic sites of C transformation and emission. Subtropical peat samples were collected from the "Lox3" core (26.597°N, 80.357°W) in October 2015 from the Loxahatchee National Wildlife Refuge, Florida. This site is an inundated mesotrophic peat marsh in the northern Everglades, with 50–100 cm of standing water. The marsh is characterized by *Cladium jamaicense* sedges and the Lox3 core was taken within 10 m of tree islands [27]. Boreal peat samples were collected from the "T3F" core (47.5063°N, 93.4527°W) from S1 Bog in July 2012 from the Marcell Experimental Forest, near Grand Rapids, Minnesota. The precipitation-fed, ombrotrophic S1 Bog has a water table ranging from 0 to 10 cm depth and is characterized by *Sphagnum spp.* groundcover and an overstory of *Picea mariana* and *Larix laricina* trees. This bog is the site of the Spruce and Peatland Responses Under Climatic and Environmental Change (SPRUCE) Experiment (http://mnspruce.ornl.gov/), where the T3F plot serves as a

control site within the experiment; coring details are described by Hodgkins et al. [27] and geochemical data for this core are reported by Tfaily et al. [38]. Polar peat samples were collected from the "CPP" core (68.3531˚N, 19.0473˚E) collected from permafrost palsa adjacent to the rim of a palsa thermokarst collapse feature in June 2012 from Stordalen Mire, a peat plateau underlain by discontinuous permafrost near Abisko, Sweden. The core was obtained from the dry (aerobic), ombrotrophic palsa soil [27], which was identified as "PHS" in [42, 43] and described therein as a recently thawed and now waterlogged thermokarst sinkhole, surrounded by palsa. The CPP core includes the seasonally thawed active layer at the time of sampling (0–60 cm), as well as permafrost peat below it (60–75 cm).

## Analyses

Prepared peat samples were freeze-dried and finely ground. Each sample was treated as a composite sample of its entire depth horizon, as listed in **Table 1**. To ensure that the data stemming from the different analyses performed on each sample corresponds to a common sample matrix, a single 2–3 gram aliquot per sample was used for the different analyses (elemental analysis, FTIR, and Ramped PyrOx with radiocarbon ($^{14}$C) analysis). The 16 sample aliquots were placed in labeled screwtop Pyrex vials and stored in a desiccator.

**Elemental analysis.** An Elementar vario EL Cube at the NOSAMS Facility at Woods Hole Oceanographic Institution was used to determine %C, %N, and C:N ratios. Five unique masses of acetanilide standard ($C_8H_9NO$; ranging from ca. 1 to 2 mg, measured with micro-analytical balance) were folded into tin boats and used to create a calibration curve for the samples ($R^2 >$ 0.999 for both C and N). A mass of approximately 5 mg of each sample was measured using the micro-analytical balance and folded into individual tin boats. Five blanks, five standards, and the 16 peat samples were run in a single sequence on the instrument.

**Fourier Transform Infrared Spectroscopy.** Molecular functional groups were analyzed using a JASCO 6800 Fourier Transform Infrared Spectrometer at Florida State University. The instrument is equipped with an attenuated total reflectance sample chamber consisting of a monolithic diamond crystal with anti-reflection coating. We use the FTIR data processing method of Hodgkins et al. [27] to best quantify the relative abundances of carbohydrates, aromatics, and aliphatics in each sample. Carbohydrates absorb energy in the infrared spectrum in a relatively wide peak centered at ~1030 cm$^{-1}$ [27] (**S1 Fig in S1 File**). The absorbance peaks of aromatics and aliphatics both have a bimodal distribution, with peaks at ~1510, ~1615 cm$^{-1}$ for aromatics and ~2850, 2920 cm$^{-1}$ for aliphatics [27] (**S1 Fig in S1 File**). Using a custom R script (https://github.com/shodgkins/FTIRbaselines; [27]), the peaks for the three molecular compound classes were located and baseline-corrected, after which the peak heights for each compound class were normalized to the total area under the curve. Area normalization of the peaks associated with each of these functional groups allows for a more direct and quantitative comparison of molecular composition across samples than humification indices, which are determined by computing the ratio of raw peak heights (e.g. aromatics to carbohydrates or aliphatics to carbohydrates).

**Ramped Pyrolysis Oxidation.** The Ramped PyrOx method mimics the natural diagenetic maturation of OM that would take place over long timescales. The ramped pyrolysis/oxidation system at NOSAMS is a custom-built analytical system designed to thermally deconvolute OM in a controlled temperature oven and purify discrete fractions of OM for subsequent C isotope analyses. A sample is progressively heated from room temperature to temperatures as high as 1,000˚C while continuously analyzing the $CO_2$ produced from the sample combustion [46, 47]. The relative proportions of labile and more recalcitrant OM in a sample can then be estimated [47, 50].

A precisely measured quantity of peat sample, approximately 1 mg C, estimated by prior % C analyses, was transferred to a pre-combusted quartz tube, sandwiched between pre-combusted quartz wool. This quartz tube was inserted into a quartz reactor and samples were prepared in oxidation mode, with pre-mixed oxygen (3 mL/min) and helium (32 mL/min) flowing through the sample. The reactor is heated by two ovens in series. The first, which heats the sample, is programmed to have a progressively increasing temperature, ramped at a rate of approximately +5˚C/min; this oven thermally separates the $CO_2$ evolving from the C in the sample as it is oxidized. The second oven in the series is set to a constant high temperature (800˚C) and heats the braided copper, platinum, and nickel catalyst wire in the lower part of the reactor to ensure quantitative oxidation of combusted C to $CO_2$. The continuously evolving $CO_2$ is quantified using an in-line nondispersive $CO_2$ concentration analyzer (LI-COR) and monitored in real-time using a custom LabVIEW computer program. Samples were prepared until the $CO_2$ concentration in the reactor outflow reached background $CO_2$ concentration (with heated oven, <40 ppm), which occurred at a sample oven temperature of ~600˚C; with the sample oven ramp rate of 5˚C per minute, this ending temperature translates to ~2 hr of active preparation, followed by passive cool down before the next sample preparation. Several of the peat samples were heated to 1,000˚C at a faster ramp rate (~1˚C/min) and none showed evidence of any C combusting at temperatures greater than 600˚C. The Ramped PyrOx preparation included separating and preserving the $CO_2$ evolved from the sample into thermally separated aliquots or "splits" for C isotope analyses, as described in the next section.

We used the model described by Hemingway et al. [50, 51] (http://pypi.python.org/pypi/rampedpyrox) to translate the data recorded during each Ramped PyrOx preparation into results that offer powerful insight into natural decomposition processes. The model was used to generate an estimated probability density function (PDF) of the activation energy (*E*) required to break the C bonds in each sample. The inputs to the model that were continuously observed during each Ramped PyrOx sample preparation to determine the PDFs of *E* were 1) measured sample oven temperature, 2) the measured $CO_2$ concentration of the gas flowing from the sample oven, and 3) time. To quantitatively compare the differences among samples that are visually observable, qualitatively, from the varying shapes of the Gaussian curves of the PDFs, the PDFs were separated into three defined *E* categories, "low" (*E* <150 kJ/mol), "medium" ($150 \leq E \leq 175$ kJ/mol), and "high" (*E* >175 kJ/mol). These three categories of *E* were arbitrarily defined in this study, based on the full range of *E* we observed in our sample set. For each sample's PDF, the area under the Gaussian curve was computed across the three ranges of the defined *E* categories to determine what fraction of the sample's C occurred in each category.

**Carbon isotopes.** The $CO_2$ produced from the Ramped PyrOx preparations was purified, separated, and collected under vacuum using cryogens (liquid Nitrogen and dry ice-ethanol slurries) into two to four sequential splits for subsequent C isotope analyses. The thermal windows for each collected split are displayed in **S1 Table in S1 File**. The thermal windows used were determined dynamically while preparing the sample set on the Ramped PyrOx vacuum line. In sampling dynamically, two sources of information were monitored in real-time and served as the basis for ending one split and beginning the next: 1) the evolving shape of the thermograph of $CO_2$ concentration burning off the sample and 2) the amount of carbon collecting in the trap to form each individual split. To maintain thermodynamic consistency across samples, each prepared sample contained a uniform amount of C (approximately 1 mg C, determined by performing elemental analyses on each sample prior to Ramped PyrOx). The preparation of 1 mg C sample provided sufficient C for both standard-sized [14]C analyses as well as δ[13]C analyses on each of the three to four splits collected from a sample. The flame-sealed Pyrex splits containing $CO_2$ from the combusted peat samples ranged in size from 59–

411 ug C, with a mean of 126 ug C (n = 52). 40 of the 52 flame-sealed samples were prepared to graphite targets and analyzed for [14]C content at NOSAMS [52–54]. An aliquot of each flame-sealed sample was taken for a $\delta^{13}$C analysis performed at NOSAMS using a VG Prism Series II Isotope Ratio Mass Spectrometer. During the two-week Ramped PyrOx preparation of this sample set, modern and fossil sodium bicarbonate (NaHCO$_3$) internal [14]C standards were prepared in the same way as samples (ca. 1 mg C and identical reactor cleaning and Ramped PyrOx procedures). The results from the analyses of these standards were used to perform [14]C blank corrections and determine "total process" [14]C error for the combined Ramped PyrOx preparation and [14]C analysis processes [55, 56] (**S2 Table in S1 File**). Separately, at the National High Magnetic Field Laboratory, subsamples of the same 2–3 g aliquots of freeze-dried and finely ground peat used for elemental analysis, Ramped PyrOx, and FTIR preparations were combusted isothermally to purified CO$_2$ to represent bulk peat samples. The flame-sealed Pyrex tubes containing CO$_2$ were sent to NOSAMS for [14]C and $\delta^{13}$C analyses so that the isotopic signatures of each bulk peat sample could be compared to the serially combusted splits for each sample prepared via Ramped PyrOx.

## Results

### Elemental analysis

The elemental analyses of the peat samples are shown in **Fig 1** (n = 16). The C content (%C) of peat in each downcore profile increased (by as much as 7.4%), apart from the Polar peat core, where %C decreased strongly with depth to values of less than 20% C at 50–55 cm and 70–75 cm depth, representing the lowest values in the sample set. The Tropical peat had the greatest %C throughout the soil profile, with values ranging between 51 and 58% C. Subtropical and Boreal peats exhibited relatively low %C in the surface and subsurface (~45% C above 25 cm depth). The Subtropical peat showed the highest nitrogen content (%N) throughout the soil profile, with values of 2.9% N in the surface, increasing to 3.5% N at the deepest depth sampled (180–181 cm). The Tropical peat had relatively high %N in the surface layer (2.0% N; 0–1 cm), below which %N decreased to 1.2–1.3% N at depth. The Boreal peat increased in %N with depth while the Polar peat had a subsurface maximum at 30–35 cm depth.

The C:N ratios of this suite of peat samples varied widely across latitudes at different depth horizons. The C:N of the surface peat ranged from 16 (Subtropical) to 50 (Polar). The Subtropical peat had the lowest C:N throughout the soil profile, with values that were relatively uniform compared to the other sites. The Tropical peat significantly increased in C:N ratio with depth, from 26 at 0–1 cm to 44–46 at sample depths ranging from 30–191 cm. The C:N ratio of

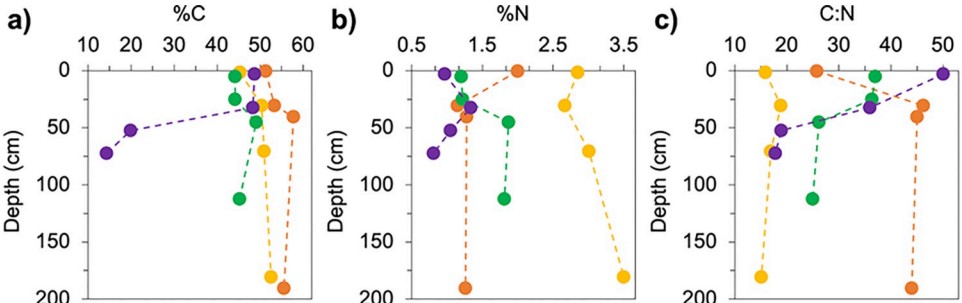

**Fig 1. Elemental analysis.** Elemental analyses were conducted to determine a) the Carbon content (%C), b) Nitrogen content (%N), and c) Carbon to Nitrogen ratios (C:N) in the Tropical (orange), Subtropical (yellow), Boreal (green), and Polar (purple) peat cores.

the Boreal and Polar peats decreased with depth and the Polar peat core had the largest range in C:N values of all sites.

**Fourier Transform Infrared Spectroscopy.** The carbohydrate, aromatic, and aliphatic data from the FTIR analyses are shown in **Fig 2** (n = 16); the raw spectra from the analyses are shown in **S1 Fig in S1 File**. The greatest abundance of carbohydrates was found in the Boreal peat, followed by the Polar, Subtropical, and Tropical peats. At all sites, there was a decrease in carbohydrates from the surface to ~ 50 cm depth, with the most apparent decline occurring in the Boreal peat profile. Below 50 cm, carbohydrates either did not change significantly or increased moderately, but not to the relatively high values found in the surface. The measured higher abundance of carbohydrates in the deepest two polar peat samples (50–55 cm and 70–75 cm) were likely altered by the presence of minerals in those samples that absorb infrared in the same region of the spectrum as carbohydrates (1030 cm$^{-1}$; [27]). The presence of silicate minerals in these two samples was detected by the presence of a small "indicator" peak in those sample's spectra, centered at 780 cm$^{-1}$ [27] (**S1 Fig in S1 File**). The carbohydrate peak heights for these mineral-contaminated samples are plotted in **Fig 2A** and denoted by open, rather than filled circles. We refrained from incorporating these values into further data analysis when assessing the agreement between the FTIR and Ramped PyrOx methods.

Aromatics were generally found to decrease with increasing latitude (**Fig 2B**). At the Tropical and Subtropical peatland sites, there was a relatively large increase in aromatics from the surface to 30–31 cm depth, below which aromatics decreased slightly, then appear to be fairly uniform downcore (to depths of 190–191 cm and 180–181 cm, respectively). The Boreal peat profile shows consistent aromatics from the surface to 20–30 cm depth, below which aromatic abundance decreased slightly and then increased dramatically in the deepest sample, 100–125 cm. The Polar peat profile shows a slight increase in aromatics in the subsurface followed by a sharp, uniform decrease in the peat sampled at 50–55 cm and 70–75 cm depth.

For aliphatics, the highest relative abundances were observed in the Tropical peat core, with the exception of the sample at 30–31 cm depth, where a sharp minimum was observed (**Fig 2C**). While the Boreal peat profile shows consistent levels of aliphatics in surface and deep peat, with a minimum at 40–50 cm depth, the Polar peat profile shows a steady decline in aliphatics with depth. Conversely, the Subtropical peat profile has relatively low aliphatics in the

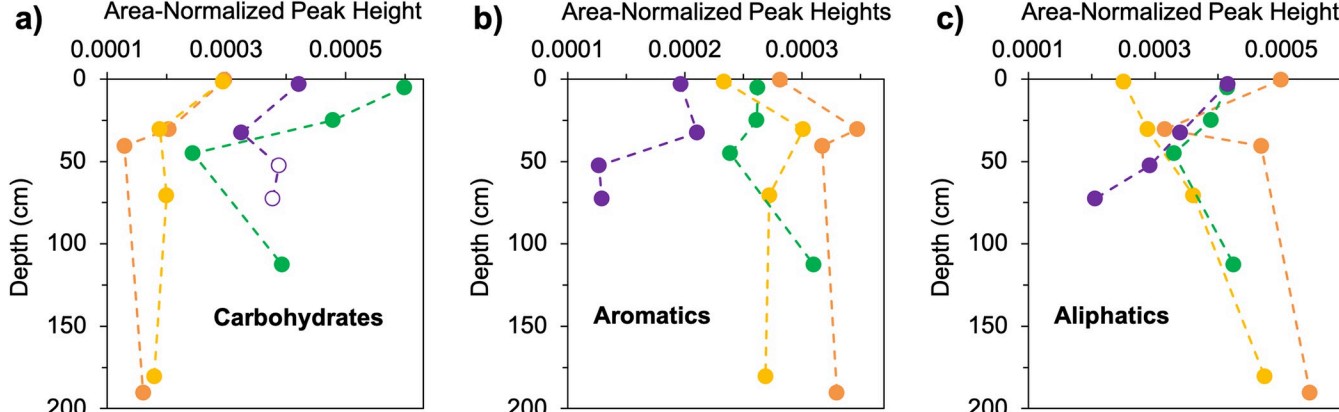

**Fig 2. FTIR depth profiles.** Depth profiles for a) Carbohydrate, b) Aromatics, and c) Aliphatics functional groups in Tropical (orange), Subtropical (yellow), Boreal (green), and Polar (purple) peat cores (n = 16). Data shown are the peak heights measured with FTIR normalized to the total area of the spectrum. In a), the open circles of the deepest two Polar peat samples indicate that these measurements were likely affected (inflated) by the presence of minerals which absorb in the same area of the IR spectrum as carbohydrates.

surface, but exhibits a steady increase down-profile. Notably, across all four climates, there are similar abundances of aliphatics in samples collected between ~20–50 cm depth.

## Ramped Pyrolysis Oxidation

**Figs 3**–**5** illustrate the observed differences in OM lability with depth in each peat core and trends across the latitudinal gradient as determined from the Ramped PyrOx approach. In **Fig 3**, the thermographs created from each Ramped PyrOx preparation are displayed, showing how $CO_2$ evolved from each sample as it was progressively heated [47]. Depth profiles for each peat core that show what proportion of C in each sample was observed to occur in the "low," "medium," and "high" defined categories of E from the model output PDFs are shown in **Fig 4**. **Fig 5** represents this data in another way by plotting the cumulative area under each p(0,E) curve as a function of E, which was calculated by finding the first derivative of its PDF. The PDFs of each sample are displayed in **S2 Fig in S1 File** and the percent of each sample occurring in each of the three defined E categories is listed in **S3 Table in S1 File**.

The depth profiles of the low E fraction (**Fig 4A**) show that the Boreal peat has the highest proportion of labile C in the upper 35 cm of the peat profiles while the greatest proportion of

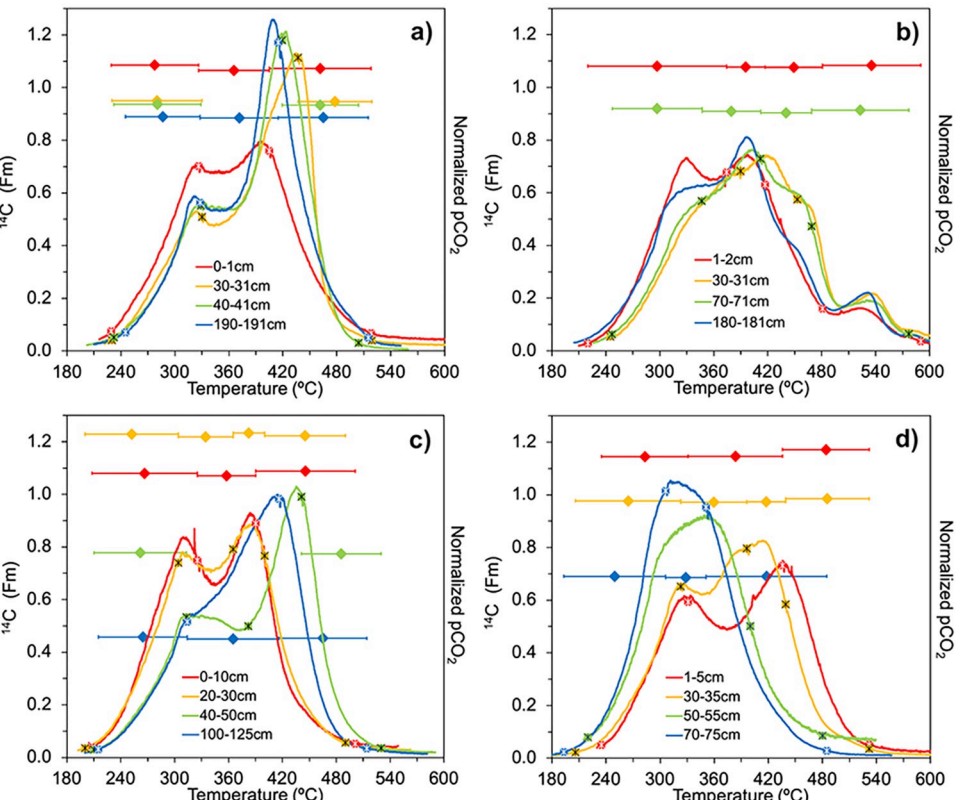

**Fig 3. Ramped PyrOx thermographs.** Thermographs show how $CO_2$ evolved from each combusting peat sample at an oven ramp rate of 5˚C per minute and the $^{14}$C measurements of the temperature-separated $CO_2$ fractions. Peat core data from a) Tropical, b) Subtropical, c) Boreal, and d) Polar climates are shown. These results are presented as function of the temperature of combustion for each serially oxidized sample. The blank-corrected $^{14}$C content of individual temperature-separated "splits" or fractions of a single sample are shown as diamonds with horizontal lines to indicate the temperature range across which each split was formed; the total process $^{14}$C error is smaller than the markers. The $CO_2$ that evolved as the sample was combusted at increasing temperature is normalized to the total amount of $CO_2$ that evolved during the sample preparation, so the area under each Gaussian curve is equal to 1. Asterisks (*) superimposed on the normalized $pCO_2$ data indicate the start/ stop of the splits that were collected for $^{14}$C analyses. The x and y axes have the same range in each subplot.

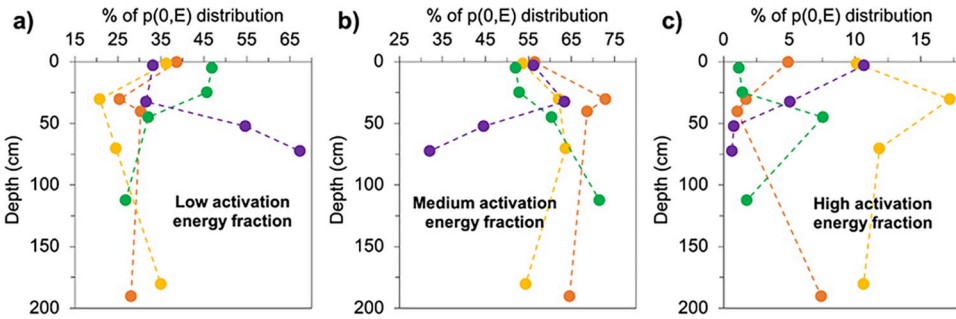

**Fig 4. Carbon reactivity depth profiles.** Depth profiles for each peat core showing what proportion of Carbon (C) in each sample was observed to occur in a) low ($E$ <150 kJ/mol), b) medium ($150 \leq E \leq 175$ kJ/mol), and c) high ($E$ >175 kJ/mol) categories of activation energy ($E$). The fraction of C occurring in each of the three defined $E$ categories was quantified by calculating the area under the curve in each $E$ category in the estimated probability density functions (**S2 Fig in S1 File**). Data is shown for the Tropical (orange), Subtropical (yellow), Boreal (green), and Polar (purple) peat cores (n = 16).

labile C overall was observed to occur with increasing depth in the Polar peat core (55% and 67% of sample occur in the low $E$ fraction at 50–55 cm and 70–75 cm depths, respectively) (**S3 Table in S1 File**). The Tropical and Subtropical cores have smaller ranges in the low $E$ fraction

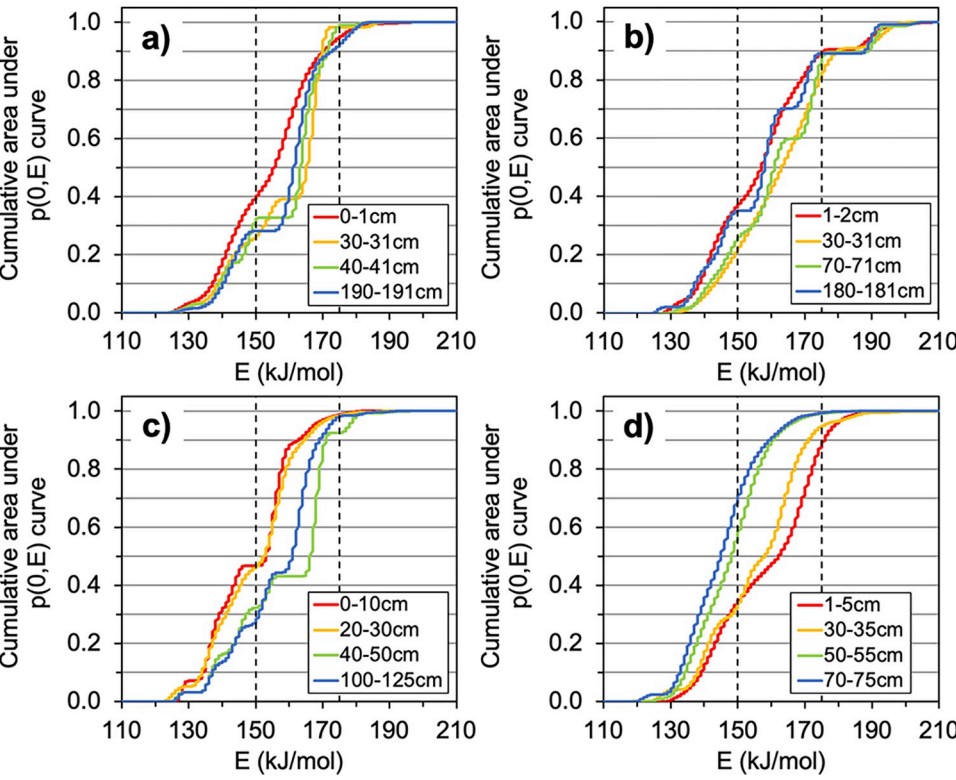

**Fig 5. First derivative of the estimated probability density function of the activation energy required to break the carbon bonds in each sample.** The cumulative area under the p(0,E) curve as a function of activation energy ($E$) for each sample depth in the a) Tropical, b) Subtropical, c) Boreal, and d) Polar peat cores. Boundaries for the three activation energy categories are represented by the vertical dashed lines. Where the data has no slope, no carbon exists at those values of $E$ and, conversely, across values of $E$ where the data has a near-vertical slope, there are relatively large amounts of carbon.

(ranges of 14% and 15%, respectively) compared to the Boreal and Polar cores (ranges of 20% and 35% ranges) (**S3 Table in S1 File**). Subsurface minima in the low $E$ fraction are observed in the Tropical and Subtropical cores at 30–31 cm depth (**Fig 4A**). The Tropical peat core has elevated medium $E$ fractions from surface to depth relative to the other climates, with a maximum of 73% of sample observed to occur in the medium $E$ fraction at 30–31 cm depth (**Fig 4B**). The proportion of peat in the medium $E$ category is similar at the surface in all four climates (52–56%) and increases from surface to subsurface. While the medium $E$ fraction increases significantly downcore in the Boreal peat (increase of 19%), the opposite trend was observed in the Polar peat, with the medium $E$ fraction halving from 30–35 cm to 70–75 cm depth. The high $E$ fraction in the sample set was small (**Fig 4C**), less than or equal to 8% in all but five of the 16 samples. Four of those five samples occurred in the Subtropical peat core, where the proportion of high $E$, recalcitrant C ranged from 10–17%, with a prominent maximum at 30–31 cm depth (**Fig 4C**).

## Carbon isotopes

The $^{14}C$ and $\delta^{13}C$ data for bulk peat samples and for analyzed $CO_2$ splits from the Ramped PyrOx preparations of peat samples (i.e. the temperature-separated fractions) are reported in **S4 Table in S1 File**. The $^{14}C$ content of the splits are plotted relative to the $^{14}C$ content of the bulk peat samples in **Fig 6**; $^{14}C$ content is reported as Fraction Modern (Fm) [57]. The $^{14}C$ content of the splits are also superimposed on the thermographs in **Fig 3**. Both figures illustrate the absence of distinct variation in $^{14}C$ content between the bulk sample and thermally separated splits for any of the peat samples; the maximum variation among splits of the same sample is 0.03 Fm and occurred in surface samples where bomb C was present. No distinct

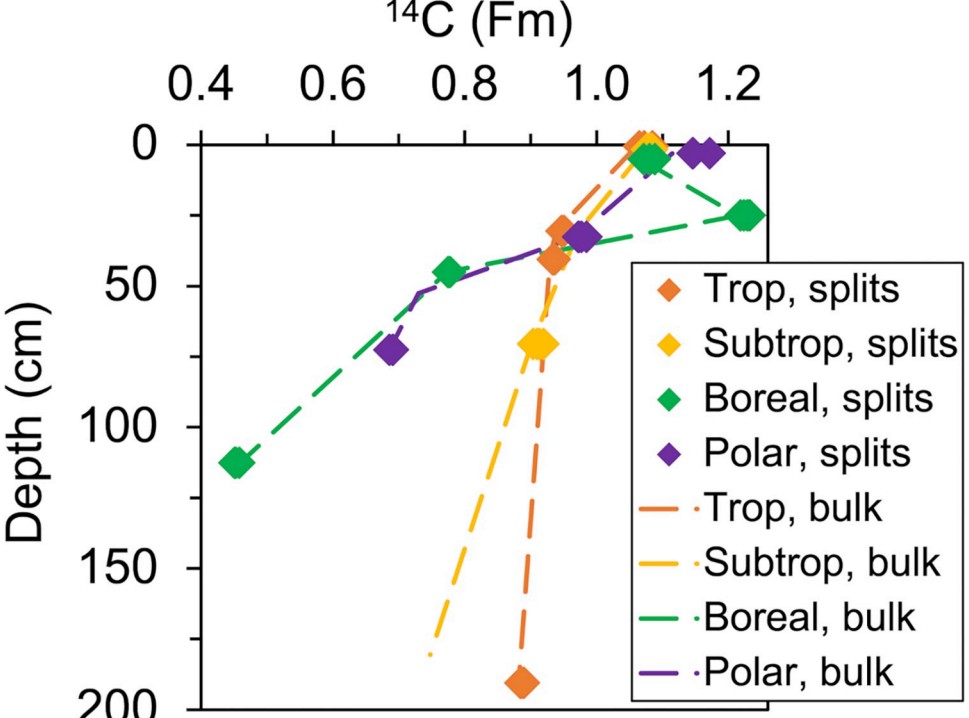

**Fig 6. Radiocarbon content of Ramped PyrOx splits compared to bulk samples.** The radiocarbon ($^{14}C$) content of the temperature- separated fractions ("splits") collected from the Ramped PyrOx method (diamonds) were observed to be equivalent to the $^{14}C$ content of the bulk sample (dashed lines). In many instances, the Ramped PyrOx splits revealed such a high homogeneity in $^{14}C$ that they appear as one marker. Error bars are smaller than the markers.

latitudinal trend for $\delta^{13}$C signatures was observed. The Tropical peat had the most depleted $\delta^{13}$C signatures, ranging from -31.4 to -28.8‰. Boreal peat had the largest range of $\delta^{13}$C values, from -30.9 to -25.9‰. The variation in $\delta^{13}$C values among splits from the same sample, was 1.1‰ on average, with a maximum variation of 1.9‰. Unlike reported $^{14}$C values, reported $\delta^{13}$C values from the Ramped PyrOx preparations could have been altered by isotope fractionation in the process of the preparations and a trend of enrichment would be expected in subsequent splits if fractionation was occurring. However, no consistent trend of enrichment as the preparation proceeded was observed (**S3 Table in S1 File**) and the $\delta^{13}$C signatures of the splits were similar to the $\delta^{13}$C signatures of the bulk samples, which were prepared isothermally on a different vacuum line. As well, the $\delta^{13}$C signatures of the fossil and modern $NaHCO_3$ standards that were prepared exhibited a trend of depletion, rather than enrichment, from first to second split (**S2B Table in S1 File**).

## Methods comparison

To evaluate H1, the hypothesis that the results from the Ramped PyrOx approach of evaluating OM quality would be consistent with that of the FTIR approach, we examined the correlations among the data stemming from these two methods using simple linear regression. Three statistically significant correlations were observed from linear regression analysis between the estimated abundance of three molecular functional groups determined from FTIR (area-normalized peak heights of carbohydrates, aromatics, and aliphatics) and the estimated fraction of each sample occurring in the three defined $E$ categories determined from Ramped PyrOx (**Fig 7, Table 2**). No significant relationship was found between aliphatics and any of the three $E$ fractions in the sample set (**Table 2**). The low $E$ fractions ($E < 150$ kJ/mol) in the sample set were positively correlated with carbohydrate content ($R^2 = 0.51$, p-value = 0.004; **Fig 7A**). The low $E$ fractions ($E < 150$ kJ/mol) were also negatively correlated with aromatics content ($R^2 = 0.58$, p-value < 0.001; **Fig 7B**). The third and final significant relationship that was observed was a strong, positive correlation between the medium $E$ fractions ($150 \leq E \leq 175$ kJ/mol) and aromatics ($R^2 = 0.69$, p-value < 0.001; **Fig 7C**).

## Discussion

### Consistency among Ramped PyrOx and FTIR methods

The three statistically significant correlations observed between the Ramped PyrOx and FTIR datasets (**Fig 7, Table 2**) support hypothesis H1 and lend confidence to conclusions made

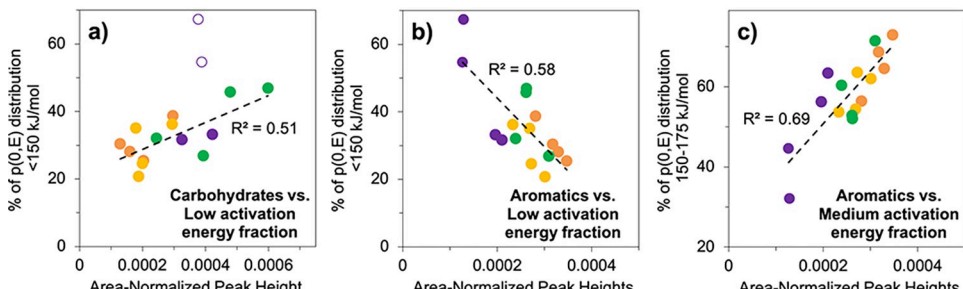

**Fig 7. Linear regressions comparing FTIR and Ramped PyrOx data.** Statistically significant relationships between normalized carbohydrate and aromatic peak heights (FTIR) and the proportion of each sample occurring in three defined categories of activation energy ($E$; Ramped PyrOx). a) Positive correlation between carbohydrates and the "low" $E$ fraction, excluding two samples affected by mineral interference (open circles), b) Negative correlation between aromatics and the "low" $E$ fraction, c) Positive correlation between aromatics and the "medium" $E$ fraction. Data is shown for the Tropical (orange), Subtropical (yellow), Boreal (green), and Polar (purple) peat cores (n = 16).

**Table 2. Correlations between FTIR spectroscopy data and Ramped PyrOx data.**

| | | FTIR: Area-normalized peak heights | | |
| --- | --- | --- | --- | --- |
| | | **Carbohydrates (n = 14)** | **Aromatics (n = 16)** | **Aliphatics (n = 16)** |
| **Ramped PyrOx: Fraction of sample in each activation energy category** | **"Low" $E < 150$ kJ/mol** | 0.51 | 0.58 | 0.12 |
| | | [0.004] | [<0.001] | [0.183] |
| | | positive | negative | – |
| | **"Medium" $150 \leq E \leq 175$ kJ/mol** | 0.25 | 0.69 | 0.18 |
| | | [0.070] | [<0.001] | [0.097] |
| | | – | positive | – |
| | **"High" $E > 175$ kJ/mol** | 0.15 | 0.024 | 0.00063 |
| | | [0.164] | [0.567] | [0.927] |
| | | – | – | – |

The first number in each cell is the $R^2$ value, to two significant digits, followed by the p-value, in brackets. The three statistically significant correlations are shaded in purple and the association between the two variables, whether positive or negative, is noted.

when employing either method to evaluate OM quality. It was expected that the low $E$ fractions ($E < 150$ kJ/mol) in the sample set would be positively correlated with carbohydrate content as observed here, because both variables are anticipated to be associated with more labile OM. The finding that the low $E$ fractions ($E < 150$ kJ/mol) were negatively correlated with aromatics content was also expected because aromatics constitute more refractory and, presumably, higher $E$ OM. This interpretation is bolstered by the third and most statistically significant relationship, a strong, positive correlation between the medium $E$ fractions ($150 \leq E \leq 175$ kJ/mol) and aromatics. Note that the small sample set studied herein and lack of core replicates presents a significant limitation of this study.

No significant relationships were observed between aliphatics and any of the $E$ fractions, which could indicate that aliphatics in these peats are distributed across a wide spectrum of activation energies. Carbohydrates did not exhibit a significant, negative correlation to the medium $E$ fraction in the way that aromatics exhibited a negative correlation with the low $E$ fraction and a positive correlation with the medium $E$ fraction. This suggests that the carbohydrates occurring in the 150–175 kJ/mol medium $E$ region are relatively stable while the low $E$ carbohydrates are reactive. The observed strong correlations, considered together, suggest that the decomposition of low $E$ carbohydrates contributes to the accumulation of medium $E$ aromatics. No significant relationships were found with any of the three compound classes measured by FTIR and the high $E$ fractions determined from Ramped PyrOx. Seven of the 16 peat samples had a high $E$ fraction that composed less than or equal to 2% of the sample, but every climate had at least one or more depths at which the high $E$ fraction was greater than 5% of the sample composition (**Figs 4**, **5**, **S3 Table in** S1 **File**).

It is compelling that there is a strong agreement among the Ramped PyrOx and FTIR approaches in this study because the two methods are very different. Both techniques have proven to be versatile tools to substantially advance our understanding of soil systems and have assisted in answering some of the pressing questions of the C cycle in recent years, but are dissimilar in terms of methodology, commercial availability, cost, labor, and required expertise. While FTIR instrumentation is a nondestructive, commercially available spectroscopic technique, Ramped PyrOx is a destructive, spectroscopic/spectrometric method with highly specialized geochemical application, and only a few custom Ramped PyrOx systems exist. An FTIR analysis of a freeze-dried, finely ground peat sample can be completed in a few minutes with minimal user training; an attenuated total reflectance accessory for FTIR is used

to enable such rapid analyses. In contrast, performing Ramped PyrOx analyses and collecting subsequent splits for $^{14}$C analyses is labor intensive, requiring a few hours of active time per sample by a person skilled in analytical chemistry and $^{14}$C vacuum line preparations, plus time for the sample oven to cool down in between samples. To allow for meaningful results and robust interpretations, both approaches require custom-developed, freely available code [created by Hemingway [51] for Ramped PyrOx (http://pypi.python.org/pypi/rampedpyrox) and by Hodgkins et al. [27] for FTIR (https://github.com/shodgkins/FTIRbaselines)] as well as quality assurance measures, namely, the analysis of reference standards. The comparison of the two techniques in this study might stimulate more cross-referencing and multi-proxy study to validate results and for selecting the adequate tool, tailored to the complexity of research questions, for a given study. With the exception of highly dynamic and heterogenous C-rich soils examined herein, additional research might support the low-resolution comparability of data from both techniques and initiate paired data augmentation.

## Latitudinal gradient in OM quality

Both the FTIR and Ramped PyrOx data suggest that peat at higher latitudes has greater OM quality than lower latitudes (**Figs 4**, **7**), a trend that has been observed in peatland studies that had significantly larger sample sets and used only FTIR [27, 30]. Again, the small sample set studied here and lack of core replicates is acknowledged as a limitation of this study and its interpretations. The two high latitude peatland sites were found to contain OM with both more easily broken C bonds (Ramped PyrOx) and greater carbohydrate content (FTIR), which indicate greater lability and susceptibility to decomposition, without considering OM protection mechanisms, compared to the lower latitude (Tropical and Subtropical) peatlands [27, 30, 32, 33, 38, 50]. The two low latitude peatland sites were observed to store OM with stronger C bonds (Ramped PyrOx) and greater aromatics content (FTIR), which indicate a greater degree of humification and/or recalcitrance relative to the higher latitude sites [27, 32, 33, 38, 50]. The deepest sample from the Boreal profile (100–125 cm depth) is somewhat of an exception to these trends, as it has a relatively high aromatics content (relative to the sample set) as well as a relatively small proportion of OM in the low *E* category. However, this sample also has a relatively high carbohydrate content, which suggests lability, so it is evident that some samples are more ambiguous in terms of OM quality and that neither Ramped PyrOx nor FTIR alone can exhaustively determine all dimensions of it.

The extent to which the latitudinal gradient in peat OM quality is observed from both the FTIR and Ramped PyrOx approaches (**Figs 2–5**) is notably not observed from the elemental analysis (C:N) data (**Fig 1**). C:N is a metric commonly used to estimate OM quality and degree of humification, with greater C:N related to greater OM quality/ lability and a lesser degree of humification [38, 58, 59]. C:N is not only driven by the degree of microbial transformation, but also by the C:N ratio of the input vegetation, atmospheric deposition, and nutrient transport via groundwater [60, 61]. Based on its C:N ratios, the Tropical peat would appear to have relatively high lability from the subsurface to depth relative to the other sites, which does not agree well with the relatively low lability inferred from both the FTIR and Ramped PyrOx analyses. The Subtropical peat has uniformly low lability relative to other cores according to its C: N ratios, which is quite consistent with what can be inferred from the FTIR and Ramped PyrOx datasets. The C:N results of the Boreal peat match the trends of the Ramped PyrOx and FTIR results fairly well, with all datasets indicating higher quality OM in the surface and subsurface samples relative to samples at depth and relative to samples of similar depths in the lower latitudes. The C:N data for the Polar (permafrost) peatland suggest that OM degrades in quality with depth and is lower quality at depth relative to Boreal and Tropical peat. In sharp

contrast, the Ramped PyrOx and FTIR results of substantial low *E* fractions (the largest in the sample set) in the Polar peat and very low aromatics content (the lowest in the sample set) assert the opposite trends of the C:N data. While the FTIR and Ramped PyrOx results suggest that the deepest peat samples from the Polar site (50–55 and 70–75 cm) have the greatest proportion of easily degradable C in the sample set, FTIR also showed that the OM in these two samples are protected by minerals [22]. The relatively low organic C content of these samples (%C; **Fig 1**) is consistent with the presence of minerals. Additionally, the 50–55 cm and 70–75 cm Polar peat samples are also the most protected by physical protection mechanisms (ice), with the deeper of the two samples collected below the base of the seasonally thawed active layer (60 cm).

### Peatlands as carbon storehouses across millennia

In peatlands, as in other soils, surface inputs are the source of labile OM. Labile molecules are preferentially consumed by microorganisms and molecules that are not consumed are buried at a rate determined by the input of new material at the surface. Physical, biochemical, and physiochemical preservation mechanisms (e.g. water, cold temperatures, ice, phenolic compounds, minerals) act to inhibit the consumption of OM. The FTIR and Ramped PyrOx data from the different peatlands analyzed in this study support the idea that the subsurface is indeed a sink for labile OM and a repository of recalcitrant OM, as demonstrated by the observed trends from the surface to subsurface of decreasing carbohydrates and low *E* fractions and increasing aromatics and medium *E* fractions (**Figs 2**, **4**). In contrast, the C:N analyses did not indicate any consistent surface to subsurface trend across the four sites. The varying quality of peat stored at depth at different latitudes, as determined from FTIR and Ramped PyrOx analyses, demonstrates the power of multiple preservation mechanisms operating in tandem that have preserved peatland OM for millennia.

It is useful, given outstanding questions about future OM degradation and warming in cooler climate peatlands to consider new information offered from the Ramped PyrOx approach about how peat OM is stored in warm climates. In the Tropical core, across a [14]C age gradient in the bulk peat of 590 to 1,010 yr B.P. from 40 to 190 cm depth, the low *E* fraction decreased by just 2% while in the Subtropical core, across a [14]C age gradient in the bulk peat of 225 to 2,340 yr B.P. from 30 to 180 cm depth, the low *E* fraction increased by 14% (**Figs 4A**, **6**, **S3**, **S4 Tables in** S1 **File**). These warm climate peatlands continue to store labile ancient C while they are undisturbed from development and remain intact as wetland ecosystems.

All four Subtropical samples from Loxahatchee NWR had substantial high *E* fractions ranging from 10–17% (**S3 Table in** S1 **File**), much of which is associated with "late" peaks observed during Ramped PyrOx preparation centered at approximately 530˚C (**Fig 3B**). Activation energy ranging from 190–205 kJ/mol would be required to break the C bonds in the most recalcitrant, high *E* fractions of these samples, whereas none of the samples from the other cores had an appreciable fraction of material in that range of *E* (**S2 Fig**, **S3 Table in** S1 **File**). These most recalcitrant fractions in the Subtropical core from Loxahatchee NWR, FL, USA, are suggestive of OM that has, through incomplete combustion, as from wildfire, been converted to pyrogenic OM. Pyrogenic OM is widely distributed in the environment and composes a significant portion of peatlands [33, 62]. According to pre-settlement fire regime reconstruction, Loxahatchee NWR had a natural fire frequency ranging from 7–25 years [63, 64]. Fires, by altering OM, have a lasting effect on OM quality, as observed in the Subtropical core. Flanagan et al. [63] demonstrated that thermal alteration by low-severity fire in peatlands can create a slower-cycling pool of OM through the creation of hydrophobic or aromatic layers on the surface of OM aggregates, consistent with the findings of Leifeld et al. [62].

Contrary to H2, the hypothesis that differently aged fractions would be observed in the Ramped PyrOx prepations of individual peat samples, the splits of each sample were all similar in age (**Fig 3**, **S4 Table in** S1 **File**). While there are large differences in $^{14}$C content downcore at each peatland site ($^{14}$C age ranges from >Modern to 6,400 yr B.P in the Boreal peat core, for example), there were no significant differences in $^{14}$C content among the splits at any one sample depth horizon nor between the $^{14}$C content of the collected splits for any sample and the bulk $^{14}$C content of that sample, prepared by combusting peat samples to $CO_2$ isothermally (**Figs 3**, 6, **S4 Table in** S1 **File**). It is evident that, at the resolution the samples were prepared on the Ramped PyrOx vacuum line (typically 50–110°C (**S1**, **S4 Tables in** S1 **File**), which translates to 10–22 min increments at the 5°C per minute ramp rate), there is no apparent variability in OM $^{14}$C content among splits or indication for differently aged components within the sample matrices. These observations are notably different from those of many other investigations using Ramped PyrOx in various settings that have isolated differently aged splits within individual samples of OM [22, 47, 65–69].

The homogenous Ramped PyrOx $^{14}$C split results from the peat samples in this study suggest that the observed fractions of differing C reactivity were all formed at the surface and buried as they aged, and that there was little to no age variation of OM input within the resolution achieved by the Ramped PyrOx preparations. The degradation that occurred fractionated the peat, consuming more readily available carbohydrates and low *E* compounds, and leaving aromatics behind, but there was not fractionation in $^{14}$C because, at any given depth, this material was all of the same $^{14}$C content. The homogeneity in $^{14}$C content of splits among labile to refractory OM in the same sample was likely driven by steady biological activity, microbial and/or peat growth. We suggest that this may be a common signal for organic soils formed authigenically, with minimal detrital inputs of differing ages, and we would not necessarily recommend adding the dimension of $^{14}$C analyses of Ramped PyrOx splits to future investigations of peatlands for this reason. It is also possible that refractory OM in the subsurface peat is slowly generated by decomposition of labile OM. Similarly aged fractions prepared with Ramped PyrOx have also been observed in New England salt marsh and pond sediments [70], though there was greater inter-sample age variation observed by Luk et al. [70] (a maximum of 0.11 Fm variation among fractions, compared to 0.027 Fm in this study; **S4 Table in** S1 **File**), which could be expected from a dynamic estuarine setting with more diverse inputs of OM.

## Conclusion

This study, which used different methods to evaluate peat OM quality and the variations in peat quality with latitude, contributes to our knowledge of the peatland C storehouse. We sought to quantify variability in peatland OM quality beyond what can be determined from % C and %N elemental analyses. A new combination of methods, FTIR and Ramped PyrOx, was applied to peatland cores from different climates to assess peat OM quality in terms of the abundance of different functional groups in a sample and the *E* required to break its C bonds. Our hypothesis that the data resulting from the Ramped PyrOx technique would correlate well with that of FTIR was confirmed. The agreement in the results of these two completely different methods for determining OM quality further strengthens the body of evidence that suggests a latitudinal gradient in the OM quality of peatlands exists [27, 30]. The two techniques were found to offer complementary information, though both methods yield meaningful results when used independently. Studies investigating OM quality using FTIR and/or Ramped PyrOx, ideally with multiple cores at each site to evaluate intra-site heterogeneity, would be further complemented by the addition of molecular microbiology to identify the microbial richness and composition with depth.

## Supporting information

**S1 File. Supporting information for peatland organic matter quality varies with latitude as suggested by combination of FTIR and Ramped Pyrolysis Oxidation.** Contains S1, S2 Figs and S1-S4 Tables.
(DOCX)

## Acknowledgments

All $^{14}$C measurements were made at the National Ocean Sciences Accelerator Mass Spectrometer Facility (NOSAMS). KJS performed the ramped pyrolysis oxidation preparations at NOSAMS and thanks the Sample Preparation Laboratory research technicians for their expert support during her multi-week visit. KJS prepared the samples for elemental analysis, which was performed by Kalina Gospodinova at NOSAMS. Samantha Bosman performed the bulk $^{14}$C preparations of the sample set at the National High Magnetic Field Laboratory. Mackenzie Baysinger, Morgan Morrow, and Angela Seibert conducted the Fourier transform infrared spectroscopy sample and reference standard analyses at Florida State University. We thank Alison Hoyt, Charles Harvey, Alexander Cobb, Suzanne Hodgkins, Curtis Richardson, Hongjun Wang, Paul Hanson, Joel Kostka, and the Isogenie field team for core sampling.

## Author Contributions

**Conceptualization:** Katy J. Sparrow, Jeffrey P. Chanton.

**Data curation:** Katy J. Sparrow.

**Formal analysis:** Katy J. Sparrow, Jeffrey P. Chanton, Ulrich M. Hanke.

**Funding acquisition:** Katy J. Sparrow, Jeffrey P. Chanton, Ann P. McNichol.

**Investigation:** Katy J. Sparrow, Jeffrey P. Chanton, Ulrich M. Hanke.

**Methodology:** Katy J. Sparrow, Jeffrey P. Chanton.

**Project administration:** Katy J. Sparrow, Jeffrey P. Chanton.

**Resources:** Jeffrey P. Chanton, Mark D. Kurz.

**Software:** Katy J. Sparrow.

**Validation:** Katy J. Sparrow, Ulrich M. Hanke.

**Visualization:** Katy J. Sparrow, Jeffrey P. Chanton.

**Writing – original draft:** Katy J. Sparrow.

**Writing – review & editing:** Katy J. Sparrow, Jeffrey P. Chanton, Ulrich M. Hanke, Mark D. Kurz, Ann P. McNichol.

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
