## [Decision Letter · Decision Letter 0]

25 Jun 2024

PONE-D-24-23109Peatland organic matter quality varies with latitude as confirmed from FTIR and Ramped Pyrolysis OxidationPLOS ONE

Dear Dr. Sparrow,

Thank you for submitting your manuscript to PLOS ONE. After careful consideration, we feel that it has merit but does not fully meet PLOS ONE’s publication criteria as it currently stands. Therefore, we invite you to submit a revised version of the manuscript that addresses the points raised during the review process.

We look forward to receiving your revised manuscript.

Kind regards,

Mingming Zhang

Academic Editor

PLOS ONE

Journal Requirements:

2.In your Methods section, please provide additional information regarding the permits you obtained for the work. Please ensure you have included the full name of the authority that approved the field site access and, if no permits were required, a brief statement explaining why.

"KJS was supported by a Diversity Postdoctoral Scholar Award from the Provost’s Office at Florida State University. KJS and JPC were awarded a Radiocarbon Research Initiative by the National Ocean Sciences Accelerator Mass Spectrometer Facility. "          

Additional Editor Comments:

I think that this paper should be major revision before publication.

General comments

The authors introduce research towards use of FTIR and the Ramped PyrOx method for assessing the quality of organic matter content of peatlands situated in four different latitudes and climates. The paper is very well-written and is clearly understandable throughout. The Methodology is detailed and mostly adequately justified. Reporting of results is sequential and clear. However, I recommend some major considerations.

Firstly, the aim of this work is not well justified and feels inconsistent throughout the paper. Initially, the paper reads as work towards replicating a finding (latitude vs OM quality trends) using another method. Whilst this in and of itself is legitimate work, the justification of why this work is necessary is not clear from the outset. As the paper progresses, more emphasis is placed on how these methods can be used in combination to help build a more complete picture of peat formation and OM quality at a site, as well as an exploration of which methods are appropriate/inappropriate for particular sites or contexts. This is well explored in the Discussion and, as a scientist with an interest in peat carbon, this felt like the most interesting part of the work to me. As such, I would suggest that the authors rework their Abstract, Aim and Introduction to better reflect the content of the Discussion. I am happy for authors to disagree with my suggestion to change the scope of their aim, but whatever the authors decide going forward, I think that a reworking of the paper is necessary so that a clear and well-justified aim is a constant thread throughout the text. This would help to make the purpose and value of the work more clear.

Secondly, I would be interested to know why more cores from each site were not sampled for Ramped PyrOx analysis. The lack of replicates for each depth section at each of the sites could have granted you more statistical power to investigate how methods compare within sites at different latitudes. Much research points towards the heterogeneity of peat carbon stocks within individual peatland systems, and the inclusion of more samples could also help you to better characterise the range of values found at different latitudes. If more samples cannot be included in a revised version, the lack of replicates at each site needs to be more thoroughly addressed as a limitation in the Discussion, and how this limits any interpretations you make.

Thirdly, no figures were available to review for the paper - only supplementary figures within the PDF were supplied to reviewers. The next version of the paper should include copies of these figures so that these can be peer reviewed too - the Results and Discussion were difficult to digest without visual aids. I am unsure whether this was an issue with the PLOS ONE editorial system and their distribution of materials to reviewers, or an issue with submission at the authors’ end. Regardless, I hope this can be resolved.

I recommend that this paper is revised with major corrections based on the above feedback and my specific comments below. I hope the authors find the comments constructive to help improve the impact of their work.

Specific comments

Introduction

L44: replace “gases” with “gas”.

It is not clear what the value of Ramped PyrOx as a method is from the Introduction. Some elaboration on the benefits of the Ramped PyrOx method (both as standalone and compared to FTIR/other methods) would be useful.

Study site and methods

A description of statistical methods used for comparison between methods is needed in this section.

Discussion

The key results which pertain to the aim of this paper are being reported in the Discussion – these should be moved to the Results section, and any interpretation of these results should remain in the Discussion.

L442: replace “intense” with “intensive”.

Reviewers' comments:

Reviewer's Responses to Questions

**Comments to the Author**

1. Is the manuscript technically sound, and do the data support the conclusions?

Reviewer #1: Yes

2. Has the statistical analysis been performed appropriately and rigorously? 

Reviewer #1: Yes

3. Have the authors made all data underlying the findings in their manuscript fully available?

Reviewer #1: Yes

4. Is the manuscript presented in an intelligible fashion and written in standard English?

Reviewer #1: Yes

5. Review Comments to the Author

Reviewer #1: General comments

The authors introduce research towards use of FTIR and the Ramped PyrOx method for assessing the quality of organic matter content of peatlands situated in four different latitudes and climates. The paper is very well-written and is clearly understandable throughout. The Methodology is detailed and mostly adequately justified. Reporting of results is sequential and clear. However, I recommend some major considerations.

Firstly, the aim of this work is not well justified and feels inconsistent throughout the paper. Initially, the paper reads as work towards replicating a finding (latitude vs OM quality trends) using another method. Whilst this in and of itself is legitimate work, the justification of why this work is necessary is not clear from the outset. As the paper progresses, more emphasis is placed on how these methods can be used in combination to help build a more complete picture of peat formation and OM quality at a site, as well as an exploration of which methods are appropriate/inappropriate for particular sites or contexts. This is well explored in the Discussion and, as a scientist with an interest in peat carbon, this felt like the most interesting part of the work to me. As such, I would suggest that the authors rework their Abstract, Aim and Introduction to better reflect the content of the Discussion. I am happy for authors to disagree with my suggestion to change the scope of their aim, but whatever the authors decide going forward, I think that a reworking of the paper is necessary so that a clear and well-justified aim is a constant thread throughout the text. This would help to make the purpose and value of the work more clear.

Secondly, I would be interested to know why more cores from each site were not sampled for Ramped PyrOx analysis. The lack of replicates for each depth section at each of the sites could have granted you more statistical power to investigate how methods compare within sites at different latitudes. Much research points towards the heterogeneity of peat carbon stocks within individual peatland systems, and the inclusion of more samples could also help you to better characterise the range of values found at different latitudes. If more samples cannot be included in a revised version, the lack of replicates at each site needs to be more thoroughly addressed as a limitation in the Discussion, and how this limits any interpretations you make.

Thirdly, no figures were available to review for the paper - only supplementary figures within the PDF were supplied to reviewers. The next version of the paper should include copies of these figures so that these can be peer reviewed too - the Results and Discussion were difficult to digest without visual aids. I am unsure whether this was an issue with the PLOS ONE editorial system and their distribution of materials to reviewers, or an issue with submission at the authors’ end. Regardless, I hope this can be resolved.

I recommend that this paper is revised with major corrections based on the above feedback and my specific comments below. I hope the authors find the comments constructive to help improve the impact of their work.

Specific comments

Introduction

L44: replace “gases” with “gas”.

It is not clear what the value of Ramped PyrOx as a method is from the Introduction. Some elaboration on the benefits of the Ramped PyrOx method (both as standalone and compared to FTIR/other methods) would be useful.

Study site and methods

A description of statistical methods used for comparison between methods is needed in this section.

Discussion

The key results which pertain to the aim of this paper are being reported in the Discussion – these should be moved to the Results section, and any interpretation of these results should remain in the Discussion.

L442: replace “intense” with “intensive”.

6. PLOS authors have the option to publish the peer review history of their article (what does this mean?). If published, this will include your full peer review and any attached files.

Reviewer #1: No

---

## [Author Response · Author response to Decision Letter 0]

14 Aug 2024

Our full 'Response to Reviewers' file has been uploaded. Thank you for your expert comments, which have strengthened the manuscript.

---

## [Editor Report · Decision Letter 1]

16 Aug 2024

Peatland organic matter quality varies with latitude as suggested by combination of FTIR and Ramped Pyrolysis Oxidation

PONE-D-24-23109R1

Dear Dr. Sparrow,

We’re pleased to inform you that your manuscript has been judged scientifically suitable for publication and will be formally accepted for publication once it meets all outstanding technical requirements.

Kind regards,

Mingming Zhang

Academic Editor

PLOS ONE
---

## [Editor Report · Acceptance letter]

12 Sep 2024

PONE-D-24-23109R1 

PLOS ONE

Dear Dr. Sparrow, 

I'm pleased to inform you that your manuscript has been deemed suitable for publication in PLOS ONE. Congratulations! Your manuscript is now being handed over to our production team.

Kind regards, 

on behalf of

Dr. Mingming Zhang 

Academic Editor

PLOS ONE